# Legal Awareness and Practices of Female Genital Mutilation/Cutting (FGM/C) among United Arab Emirates Medical Practitioners

**DOI:** 10.3390/ijerph20064710

**Published:** 2023-03-07

**Authors:** Shamsa Al Awar, Kornelia Zareba, Gehan Sayed Sallam, Nawal Osman, Teodora Ucenic, Howaida Khair, Suzan Al Shdefat, Hadya Abdel Fattah, Sara Maki

**Affiliations:** 1Department of Obstetrics and Gynecology, College of Medicine and Health Sciences (CMHS), United Arab Emirates University (UAEU), Al Ain 17666, United Arab Emirates; 2Department of Obstetrics and Gynecology, Faculty of Medicine, Jordanian University of Science and Technology, Ar-Ramtha 3030, Jordan; 3Nursing Department, Fatima College of Health Sciences, Al Ain P.O. Box. 24162, United Arab Emirates

**Keywords:** FGM/C, FGM, female genital mutilation/cutting, reasons for performing FGM/C, FGM/C law, FGM/C awareness

## Abstract

Female genital mutilation/cutting (FGM/C), due to its regional occurrence, is a marginalized issue in the international arena. The aim of the study was to verify reasons for performing a procedure prohibited by international and domestic law. A cross-sectional study was conducted among nurses and doctors practicing in the UAE. The study was conducted from the 1 of January 2020 to the 31 of December 2021. The total number of recruited individuals who agreed to participate was 120, with the rate of return being 82%. About half of the participants (*n* = 59, 49.2%) have seen FGM/C patients in their UAE practice. Regarding medical staff, the total knowledge score concerning possible complications of the performed procedure was assessed at 64%, i.e., at a moderate level. None of our study participants had previously performed any type of FGM/C. However, 6.7% were willing to do it upon a mother’s or guardian’s request. About 83% of study participants stated that FGM/C should be halted internationally. Only 26.7% of the medical practitioners were aware of UAE law concerning FGM/C, while 50% had no knowledge concerning this issue. The present study reveals that cultural conditions take priority over medical knowledge, making medical practitioners inclined to accept the circumcision of girls and women. The crucial goals of future activities should focus on sensitizing society and the medical community, the need to create clear laws penalizing the practice, and the legal obligation to report the circumcision of a girl or woman.

## 1. Introduction

Female genital mutilation/cutting (FGM/C) has been defined by the World Health Organization (WHO) as “all procedures involving partial or total removal of the female external genitalia or other injury to the female genital organs for non-medical reasons” [1]. FGM/C may be classified as Type I (the excision of the prepuce with or without a partial or total excision of the clitoris); Type II (excision of the prepuce and clitoris with a partial or total excision of the labia minora); Type III (excision of a part or all of the external genitalia and the stitching/narrowing of the infundibulum); Type IV (unclassified; piercing, pricking, or an incision into the clitoris and/or labia) [2,3].

Due to its regional occurrence, female genital mutilation/cutting is a marginalized issue in the international arena. However, it must be remembered that over 90% of girls in Sub-Saharan Africa undergo the procedure. According to data reported by UNICEF in 2021, the estimated prevalence of women around the world who have been subjected to FGM/C exceeds 200 million. The procedure is performed with varying prevalence in around 31 countries, mainly in Sub-Saharan Africa, the Middle East (Egypt, Oman, Yemen, and the United Arab Emirates), and in some countries in Asia (India, Indonesia, Malaysia, Pakistan, and Sri Lanka) [4]. A study conducted by Al Awar et al., revealed that the prevalence of FGM/C in the UAE was 41.4%. Type I was most prevalent (62.8%), followed by Type II (16.6%) and Type III (5%) [5]. General globalization is the reason women subjected to FGM/C are also found in European countries. A study conducted in Spain revealed that 5.4% of medical practitioners met circumcised women in their practice [6]. The issue of FGM/C also directly affects health-care workers. Research conducted in Nigeria showed that 44% of the nurses surveyed had undergone FGM/C [7].

A collaborative prospective WHO study conducted in six African countries in a group of 28,393 women demonstrated that women who had undergone FGM/C were significantly more likely to experience adverse outcomes, especially obstetric outcomes, compared to those without FGM/C [8]. Along with biological complications, it is extremely important that to know the mental health of mutilated women is threatened [9,10]. Physiological complications are divided into those that are short term and long term. Immediate or short-term complications of FGM/C include traumatic bleeding, severe pain, genital tissue swelling, urine retention and wound infection [11,12]. Long-term complications are recognized to have an effect throughout women’s life spans. These are divided into two main areas: gynecological and obstetric. Long-term gynecological complications related to FGM/C include infection, development of scars and keloids, menstrual difficulties, urinary symptoms, and infertility [13]. According to the literature, long-term complications carry a significant risk regarding future maternal health, leading to sexual dysfunction and dyspareunia. Obstetric complications include perineal and episiotomy tears and prolonged labor. Moreover, these may obstruct assessment and treatment by health-care workers, including during intrapartum vaginal examination or in catheterization [14]. Post-traumatic stress disorder (PTSD), affective disorders, anxiety, somatization, and depression are the most common psychological complications [15].

FGM/C is mostly performed by traditional practitioners, e.g., circumcisers and birth attendants. Regarding medical staff, FGM/C procedures are most commonly performed by nurses and midwives [16]. A study conducted in Nigeria showed that nurses and midwives constituted seven out of eight medical workers asked to perform the procedure [16]. Similarly, according to a study conducted in Guinea, the procedure was carried out mostly by traditional excisers (88%) but also by midwives and nurses [17]. Onuh et al., reported that the nurses performing FGM/C (6.6%) were older women with many years of experience in the profession [7]. 

Many countries have ratified and deposited the Maputo Protocol, which promotes women’s rights, including an end to FGM/C [18]. A majority of respondents in the cited studies agreed with the view that the FGM/C procedure violates human rights [19]. This applies both to the self-determination of one’s own body and to the woman’s right to experience pleasure, which she is deprived of by removing the clitoris. Regrettably, despite existing laws prohibiting the procedures, they are not enforced in many countries. For example, the UAE Federal Decree-Law No. (4) of 2016 on Medical Liability does not define FGM/C literally as a prohibited procedure. However, Article 4, paragraph 3 clearly states that physicians have to “use the available diagnostic and treatment tools required for the case”, which can be interpreted as a ban on performing a medically unjustified procedure [20]. Moreover, Article 5, paragraph 10 says that a “physician shall not (…) take unnecessary medical actions or perform unnecessary surgeries in a patient without obtaining his/her informed consent”, and Article 15 states that “no action or intervention may be made for the purpose of regulating reproduction unless upon request or consent of the two spouses”. Obviously, it is impossible to obtain informed consent from a child. Furthermore, this procedure is associated with complications that will occur until the end of life. Article 6, paragraph 2 defines medical error as “non-compliance with the recognized professional and medical principles”. The FGM/C procedure stands in opposition to the worldwide principle of nonmaleficence, termed primum non nocere in Latin. Recommendations of the international conference on Zero Tolerance to FGM, held in Addis Ababa in February 2003, unanimously state the need to stop the procedure [21]. However, despite potentially high penalties, the procedure is still widely practiced in Guinea and other African countries, while the percentage of women subjected to it worldwide has decreased only slightly [17]. This clearly indicates that social acceptance still exists, even among medical personnel.

FGM/C has extremely deep roots in the cultures, social values, and beliefs of the affected communities, including the UAE [22]. A majority of people and patients in these communities believe that FGM/C is intended to protect women’s chastity by reducing the libido, thereby safeguarding family honor [23]. In addition, it is taken as an expression of love for daughters and the desire to provide them with a proper future free of social ostracism. Additionally, these communities offer various reasons for practicing FGM/C, e.g., religion. Muslims widely believe that girls and women must be subjected to FGM/C [24]. In a study conducted in the Gambia, some respondents reported that Islam required FGM [25]. However, regarding the religion of Islam, there is no statement in the Quran authorizing or recommending that Muslims perform the FGM/C procedure. 

A study conducted by our team among 1035 UAE residents in 2016 and 2017 demonstrated that 36.7% of FGM/C procedures were performed by medical staff—most commonly in private hospitals [5]. Based on this finding and on the issue’s controversial nature, we decided to verify the reasons for performing a legally prohibited procedure. Therefore, we asked questions concerning respondents’ medical knowledge regarding the complications, their personal beliefs, and their awareness of the law in force in the country.

## 2. Materials and Methods

The cross-sectional study was conducted among nurses and doctors practicing in the UAE, both in the governmental and private sectors. The study was conducted in accordance with the Declaration of Helsinki, and the protocol was approved by the Al Ain Medical District Human Research Ethics Committee (AAMDHREC) (ERH-2015-3120 15-101).

### 2.1. Material

The original aim had been to conduct the study over a period of 6 months. However, due to extreme difficulties in recruiting participants, this was extended to 2 years (from the 1 of January 2020 to 31 of December 2021). The questionnaire was distributed to all medical professionals throughout the UAE who directly participated in women’s care, including physicians, nurses, and midwives, both in governmental and private clinics and hospitals. Due to problems with distributing paper questionnaires among health care facilitators, a personal-contact approach was made with most facilities. The issue was discussed with participants in person on a one-to-one basis. The total number of recruited individuals who agreed to participate was 120, with the rate of return being 82%, as the questionnaire was distributed through MOH, DHA, and ADH as per communication with UAE medical-licensing agencies. Due to the cross-sectional character of the study, there was no control group.

### 2.2. Questionnaire

The questionnaire was developed by the present authors, based on updated and reviewed literature, and then applied to UAE social and cultural implications. It was piloted for cultural validity and reliability among College of Medicine and Health Sciences OB/Gyn Department faculties, research nurses, and staff. Amendments were based on feedback from the pilot subjects. These piloted survey participants were then excluded from the study group. The questionnaire was divided into three main parts. Part I covered sociodemographic data, including the age, nationality, gender, religion, medical designation, and workplace. Part II covered knowledge of FGM/C in the UAE, as well as practices encountered by the participants. It contained 13 questions concerning such issues as medical knowledge, knowledge of UAE law, personal experiences, and practices. Part III covered the UAE medical liability law and the awareness of the participant. It was composed of a total of six questions. 

### 2.3. Statistics

Initially, a descriptive analysis was carried out of the variables under study and the assessment of associations between them. A majority of outcome measures were based on dichotomous scale coding. Socio-demographic variables of the study population were presented as numbers and percentages. The following statistical tools were implemented: the comparison of independent continuous variables, the Student’s *t*-test, the analysis of variance (ANOVA) with post-hoc multiple comparisons (Scheffe Model), and the Pearson’s correlation coefficient (r). The results were considered statistically significant at *p*-value ≤ 0.05.

Data entry was checked, coded, and analyzed using the Statistical Packages of Social Sciences (SPSS) version 19 software (IBM, 2010).

## 3. Results

### 3.1. Socio-Demographic Factors

The total number of respondents (participants) was *n* = 120, and the highest age group was 41–50 years (*n* = 42, 35%). Regarding their nationality, Indian participants ranked first (*n* = 35, 29.1%), followed by the Sudanese (*n* = 20, 16.8%). Regarding national group or class, most respondents were from Asia (*n* = 52, 43.3%). Muslims constituted a majority of study participants (*n* = 79, 65.8%) (Table 1). 

Almost all participants were women (*n* = 118, 98.8%). About half of them were consultants (*n* = 54, 45%), while the lowest designation group included medical officers and resident doctors (*n* = 9, 7.5%). Most of the study participants were from government hospitals (41.7%) and private hospitals (34.2%) (Table 1).

### 3.2. FGM/C Professional Knowledge

Only 3.3% of the participants considered FGM/C as an easy procedure without complications. According to participants, severe pain (80%), hemorrhage (70%), urine retention (67.5%), wound infection (66.7%), urinary-tract infection (65%), fever (65%), and septicemia (57.5%) were the most possible immediate complications of FGM/C (Figure 1). 

Regarding long-term consequences, more than two-thirds of the study participants reported having such complications during childbirth, with hemorrhage, urine retention, wound infection, urinary tract infection, fever, and septicemia also being reported. (Figure 2).

The knowledge domain included 10 items for possible immediate and 10% for possible long-term complications with (yes/no) answers. All the questions were weighted across the total score out of 100%. Total knowledge score was established at 64%, which corresponded to a moderate level.

### 3.3. FGM/C Experiences in the UAE

About half of the participants (*n* = 59, 49.2%) reported that some patients in the UAE (nationals and/or residents) had undergone FGM/C, and 18 participants (15%) did not know any. Type I (*n* = 55, 45.8%) was the most common type of FGM/C in the UAE. Moreover, cases of type II (*n* = 17), type III (*n* = 9), and 1 person with type 4 were also reported.

None of our study participants had performed any type of FGM/C. Only 13.3% of the participants had faced complicated cases, with the most complicated cases, including difficult labor, resulting in an extensive tear (third-degree tear) and difficult obstetric examination.

The most common nationalities asking for FGM/C were the Sudanese (15%), Somalian (7.5%), other African nationalities (6.6%), and other non-African nationalities (10.8%).

### 3.4. FGM/C in Personal Opinions

About 29% of the participants agreed that FGM/C was a culturally accepted issue, while only 7.5% agreed that FGM/C was religiously recommended (Table 2. Regarding the participants’ attitudes, none would agree to perform FGM/C based on a patient’s request (Table 2). However, 6.7% would consider performing the procedure upon the request of a mother or guardian. None of the participants would agree to perform FGM/C Type III or IV based on the request of a mother or guardian. About 28% of participants would use anesthesia if they were performing a FGM/C procedure. The most common minimum age (in years) at which the doctors would be willing to perform the procedure was 2, 9, or 14 years old. Conversely, almost all participants (about 98%) would not agree or give consent to circumcise their daughter. Finally, none of the study participants would be willing to advertise themselves as an FGM/C practitioner.

The attitude domain included five items with (yes/no) answers, and all the questions were weighted across the total score out of 100%. The total score for statements against FGM/C was 90.3%, which was considered as a positive attitude based on a cutoff point of 50%.

Regarding participants’ opinions according to their nationality, there was no statistically significant association (*p*-value > 0.05). However, Europeans reported the highest proportion for “Internationally abandon” (90.9%), while Africans had the highest proportion for “Culturally accepted” (13.3%) and for “Religiously recommended” (6.7%). Moreover, 46.7% of Africans reported that the procedure is “Religiously recommended” (*p*-value < 0.001).

### 3.5. FGM/C UAE Law Awareness 

Regarding participants’ knowledge about FGM/C, about 83% of them considered FGM/C a procedure that should be abandoned internationally. Only 26.7% already knew that the UAE had a federal law concerning FGM/C, while 50% had no knowledge on this issue. A majority of participants thought that FGM/C should not be an accepted medical practice in the UAE. Moreover, a majority (about 94%) discouraged performing FGM/C in the UAE. Only one participant agreed that FGM/C should be an accepted medical practice in the UAE. About half of the study participants (47.5%) agreed that they should be reported if performing FGM/C, while almost 36% stated that they would not perform it under any circumstances. Total participants’ awareness concerning the UAE Medical Liability Law was 74.3%, which means that about three-quarters of our study participants knew, respected, and accepted medical-liability laws regarding FGM/C in the UAE.

The Pearson’s correlation value showed that there was a statistically significant positive correlation between knowledge and attitude/practice of the study participant, with r = 0.662 and *p*-value = 0.001. The ANOVA analysis (Scheffe Model) and independent sample *t*-test revealed a statistically significant difference in the mean score of knowledge and the nationality group, with f-test = 3.64 and *p*-value = 0.015. A significant difference occurred between Arabian and Asian groups, with the *p*-value of 0.008. However, no significant differences were noted in the mean knowledge score or the age group, sex, religion, place of work, and designation, with the *p*-value over 0.05 (Table 3).

## 4. Discussion

The presented research revealed that about half of the participants (*n* = 59, 49.2%) had FGM/C patients in their UAE practice (nationals and/or residents). The most common nationalities asking for FGM were the Sudanese (15%), Somalian (7.5%), and other non-African nationalities (10.8%). None of our study participants had performed any type of FGM/C. However, 6.7% would consider doing so upon the request of a mother or guardian. About 83% of study participants considered FGM/C to be a procedure that should be abandoned internationally, and a majority of participants thought that FGM/C should not be an accepted medical procedure in the UAE. However, only 26.7% knew that the UAE has a law concerning FGM/C, and 50% had no knowledge concerning this issue. A statistically significant positive correlation occurred between the knowledge and attitude/practice of the study participants. 

The study demonstrated a lower percentage of FGM/C compared to the previous study [5]. This may result from the character of the surveyed group. The current study was conducted among medical practitioners, and the previous one was conducted among residents in the local population. However, the results are significantly lower than in other regions of the Muslim world. A study conducted to assess attitudes of Guinea health care providers toward FGM/C revealed that Guinea had the highest prevalence of female genital mutilation (95%) [17]. Percentages also vary among various communities living in a country. A study conducted in the Gambia revealed the lowest percentage in the Wolof population (12.4%) and the highest among the Mandinka (85%) [25].

In the present study, the level of medical knowledge regarding the procedure and complications occurring after FGM/C was determined as average. A survey conducted among midwives in Eastern Sudan showed that 80.9% of them practiced FGM/C at some point in their lives, but only 7% (11/157) could identify four types of FGM [26]. Decreased sexual pleasure (64.3%, 101/157) was the most commonly reported complication of the procedure in the study. A study from the Gambia showed that 40.9% of health practitioners had encountered complications related to the FGM/C procedure [25]. Their level of knowledge was generally higher. A study conducted in Nigeria revealed that the level of knowledge concerning FGM/C complications was very high. Therefore, 90% of those medical practitioners stated that FGM/C was not a good practice [16]. A study carried out in Spain showed that only 10.7% of respondents had appropriate knowledge in the area of FGM/C types, and 33.9% knew the applicable laws in the country. This might be due to the fact that the percentage of people subjected to FGM/C was low, and only 5.4% of respondents had met them in their practice [6]. A study conducted in Alexandria by Mostafa et al., showed that medical students had little knowledge of the FGM/C procedure and of complications, and a significant percentage (52%) supported the legitimacy of its implementation. Similarly, a study by Afifi et al., demonstrated a correlation between the level of education of medical staff and the frequency of FGM/C performance [27]. Sometimes, the level of knowledge among staff was so negligible that respondents did not know whether they had been subjected to the procedure, as was the case in a study carried out in Nigeria, with 11.5% of respondents not being able to determine this [7]. 

The rationale for the medicalization of FGM/C has been raised by numerous authors. This would certainly contribute to reducing the number of complications and to increasing the possibility of performing the procedure under anesthesia. On the other hand, however, making FGM/C a permitted procedure would legitimize it. According to a study carried out in the Gambia, a tendency to medicalize the procedure was observed. This may have resulted from the many complications being observed in the country [25]. A study conducted there in rural environments showed that 42.5% of health facilitators stated that FGM/C should still be performed, and 47.2% would have their daughters undergo the procedure [25]. The study also demonstrated male respondents’ greater tendency to support the procedure. Medical practitioners decide to perform the procedure mainly for reasons of worldview, as they had been brought up in the same tradition. Moreover, traditions were placed above medical knowledge. Financial gratification and being regarded as a “local celebrity” in the community often play an important role. This group of medical practitioners claims that the medicalization of the procedure would reduce suffering and the number of complications [28]. The present study showed that none of the study participants would willingly advertise themselves as FGM/C performers, yet high support for the medicalization of the procedure was also observed in Egypt, mainly from general support for it in the medical environment, where a majority of medical practitioners still support the procedure [29,30]. A study conducted in Ethiopia showed that 49% of medical practitioners believe FGM/C should still be practiced, and 13.5% would circumcise their daughters [19]. According to that study’s authors, 40.5% of respondents reported that they had performed FGM/C procedures on girls younger than five. Similar results were reported in studies from Egypt and western countries [25,29]. However, in most countries, medical personnel are in favor of banning FGM/C. A study conducted in Guinea demonstrated that a majority of the surveyed doctors were against FGM and its medicalization. A total of 89% of respondents there stated that the procedure violated women’s rights, and 81% stated that it should be legally penalized [17]. 

An important issue is the law in force in a country, how it is administered, and the possibility of enforcement. Medical law in the UAE does not provide any penalty for carrying out FGM/C procedures. Notably, medical law only applies to persons performing as medical professionals. Traditional circumcisers are not licensed to perform such procedures, and the law imposes extremely severe penalties. Despite this threat of punishment, the procedures continue to be performed. In Guinea, as with the UAE, despite national law prohibiting FGM/C, it is not enforced [17]. This is mainly due to the lack of reports from medical practitioners who found ritual damage to a patient’s body. It may also result from the lack of awareness of the applicable law, though it is known that ignorance of a law is no exemption from having to obey it. An obligation to report evidence of circumcision could contribute to the cessation of the procedure, which, in many countries, could also apply in the case of rape of a minor. 

While a study conducted in Eastern Sudan revealed that 74.5% of the surveyed midwives claimed FGM/C to be a lawful practice [26], in the UAE, law-enforcement possibilities leave no doubt. Therefore, receiving information about legal violations remains the weakest link, along with the extent of penalties. The low occurrence of rape reported in the UAE is the clearest proof that effective laws and severe penalties constitute the firmest factors in preventing crimes. Most countries have introduced FGM/C legislation restricting implementation of the procedure. Interestingly, in Guinea, where the procedure is still widely performed, most medical practitioners surveyed were in favor of ending the procedures not due to moral and cultural reasons but to the law now in force there and to medical complications that occur [17]. The problem does not only concern countries where FGM/C is practiced for cultural reasons [31]. A study conducted in Spain revealed that only 31.7% of medical practitioners, after finding lesions resulting from FGM/C, decided to address the issue in the consulting room, and 23.8% asked about children in the family who could be subjected to the procedure in the future [6]. None of the respondents reported this fact to the authorities. Therefore, sensitization within the medical environment, along with a requirement to report evidence, appears to be the weakest link. In standard FGM/C procedures, young children are mutilated in most cases.

One important problem this topic brings to the fore is the legitimacy of performing procedures that are medically unjustified. Most people from the medical community state that, for this reason, FGM/C procedures violate medical ethics [17]. A crucial assumption in the medical profession is what is referred to as the primum non nocere principle, i.e., not harming the patient. An overwhelming majority of FGM/C procedures involve more pain associated with the procedure than with the recovery process. They also cause long-term complications that the patient may experience with each act of sexual intercourse. Therefore, performing medically unjustified FGM/C procedures contradicts initial assumptions of the medical profession and constitutes malpractice. A survey conducted in Guinea showed that 29% of medical workers claimed that circumcised women were more faithful in marriage than those who were not circumcised. According to a survey conducted in Ethiopia, 26.7% of respondents stated that women who had not undergone FGM/C should be discriminated against [19]. This remains in opposition to basic principles doctors are required to follow regarding equal treatment. Moreover, 56.4% of respondents stated that the tradition could not be eradicated, and 11% claimed that it was a good practice [17]. 

FGM supporters may suggest that plastic surgery or piercing also have no medical justification and may cause pain and complications. It must be emphasized that each FGM/C procedure is irreversible, in consideration of possible medical complications and the extent of damage to the person’s body, even should psychological aspects be set to the side. We do not have the technical possibilities, at present, to transplant or restore the clitoris with its blood supply and innervation. Furthermore, plastic surgery procedures are performed only after obtaining informed consent. It should also be noted that sex-reassignment surgeries for transsexualism are already prohibited in most countries where FGM/C procedures are performed. Thus, certain inconsistencies are in place, in prohibiting the former procedures but tolerating the latter, when attempting to understand the ethical issues of these medical procedures. 

The FGM/C procedure is deeply rooted in many cultures. In a survey conducted in Ethiopia, few medical practitioners considered FGM/C a religious practice (17.8%), and a majority stated it was a cultural practice (78.2%) [19]. Similarly, a Nigerian study revealed that 96.6% of respondents reported reasons ranging from cultural (96.6%), religious (12.7%), and related to beautification (3.4%) and hygiene (2.5%) [16] as the main causes for FGM/C practices. Therefore, it seems reasonable to return to this practice’s roots, as knowledge of its genesis gives the opportunity to argue in this area. According to numerous interpretations, the FGM/C procedure was introduced to control women’s sexuality and ensure their so-called purity [32]. Advocates of the procedure claimed that it protected femininity and guaranteed family honor, as well as prospects of getting married [33]. Another frequently raised issue was that FGM/C reduced the risk of promiscuity (38%) and prostitution [17,19]. This belief was more characteristic of nurses (75%) and auxiliary health providers (63%) [17]. 

FGM/C proponents also suggest that it prevents social stigma and supports social identification among young women. Broad education of a given society in the area of sexual life may constitute a solution when facing abnormal mental patterns that are being perpetuated in the social consciousness [34]. In the case of FGM/C practices, a more humane solution than the ritualistic mutilation of women may well be the education of men in terms of living according to religiously recognized principles and not indulging in prostitution. Therefore, refusing social consent for extramarital relationships and prostitution, in keeping with the teachings of the Quran, can provide a mutually beneficial method for a society. The custom of FGM/C procedures is culturally distinctive, and the religion of Islam does not unambiguously indicate that the procedure should be practiced. A characteristic feature of culture is its evolution over time. Therefore, awareness and broadening public support for ending this procedure should be promoted. 

A change in social awareness in the medical environment is also necessary. A major objective in the activities of medical communities should be social and environmental actions, especially among ethnic minorities, among whom the incidence of FGM/C procedures is very high. In practice, a law in force in a country that is not connected with widespread social awareness of its existence and support is a dead law. A study conducted in Guinea showed that health care providers were looking for training in the treatment of FGM/C complications and in the recognition of damage, as well as to improve skills in refusing to perform the procedure. Medical providers are the strongest link who can contribute to ending this practice. Especially in the education programs for gynecologists, pediatricians, nurses, and midwives, the subject of FGM/C procedure and applicable law should be included. Awareness of cultural traditions in specific countries will help to distinguish risk groups that should be addressed. They should be made aware that FGM/C is a criminal practice associated with numerous medical complications. Medical practitioners prepared with appropriate medical knowledge can provide a greater authority for the given social groups than advocates of a harmful custom. As the world becomes a global village, cases of FGM/C and its attendant health and social complications arise in many countries in addition to African ones [35]. Therefore, we must no longer marginalize the concerns of millions of women around the world.

Many voices emphasize the need for intercultural mediation in the field of medicine as well, especially in cases of cultural and ethically sensitive issues [6]. It is reasonable and responsible to create a commission consisting of legal, medical, and religious representatives tasked with establishing a consensus and formulating clear legal recommendations that can, then, be provided to and enforced among medical personnel and society at large. 

### 4.1. Strengths of the Study

The present study is the first to be conducted among the medical community in the UAE. Since there are no legal penalties for carrying out FGM/C in the UAE, there were no repercussions for complete honesty of performing or accepting the procedure. It raises an important and unavoidable subject that has, to date, remained hidden from view. It draws attention to the weakest links contributing to the continued existence of the problem both in the country and throughout the region. It then proposes adequate and feasible solutions in respect to the local culture and society overall.

### 4.2. Limitations of the Study

The initial problem we encountered during the study was the difficulty of recruiting the group of respondents; then, there were problems with its small size. Even after prolonging the study’s duration, the group remained small. The aim had been for the study to last six months, yet due to extreme difficulties in recruiting participants, it was extended to two years. This might be associated with the sensitive topic and concerns about potential legal implications. Due to the small group size, the participants may not constitute a representative group across the whole country. Based on the previous study, we recognize that the problem may affect a much larger group of women and medical practitioners. Due to the lack of professional psychological tests, the study was carried out on the basis of the present authors’ own survey.

## 5. Conclusions

The lack of widespread knowledge among the medical environment, with regards to potential complications due to FGM/C practices, may contribute to increased support for the procedure. Cultural conditions outweigh medical knowledge, making some medical practitioners inclined to accept circumcision of girls and women (including their own daughters). Educating the population in terms of sexual life and religious dogmas should change inaccurate mental patterns that predominate in the social consciousness. The most important goals for present activities should focus on the sensitization of society and the medical community, along with the need to establish clear laws penalizing FGM/C practices and requiring that any woman’s or girl’s circumcision be reported appropriately.

## Figures and Tables

**Figure 1 ijerph-20-04710-f001:**
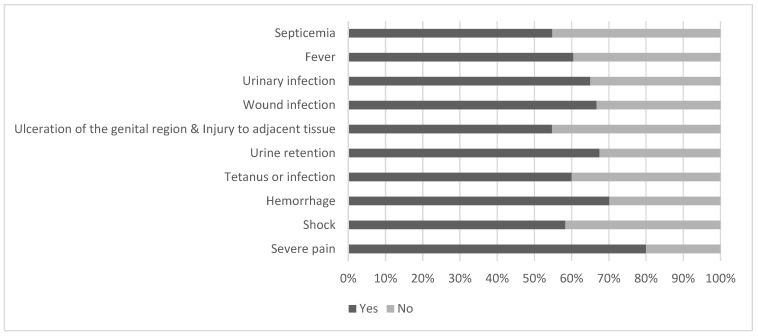
Participants’ knowledge about possible immediate complications caused by FGM in percetages (*n* = 120).

**Figure 2 ijerph-20-04710-f002:**
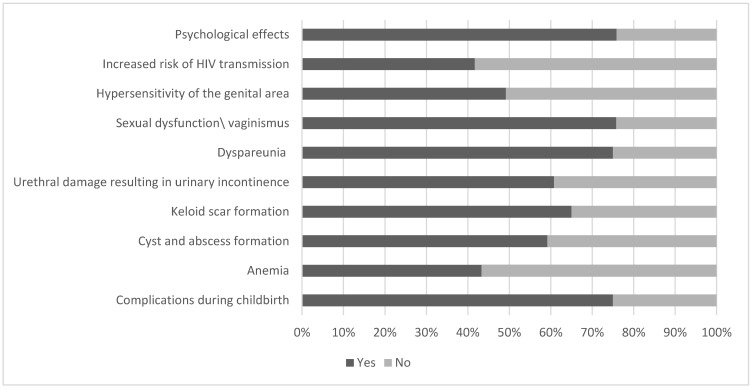
Participants’ knowledge about possible long-term complications caused by FGM in percetages (*n* = 120).

**Table 1 ijerph-20-04710-t001:** Socio-demographic characteristic of the participating group (*n* = 120).

	Frequency (*n*)	Percent (%)
**Age (Years)**		
18–30 Years	19	15.8
31–40 Years	28	23.3
41–50 Years	42	35.0
51–60 Years	31	25.8
Total	120	100.0
**Nationality**		
Afghanistan	4	3.4
Bangladesh	4	3.4
Egypt	11	9.1
India	**35**	**29.1**
Iraq	12	10.0
Pakistan	7	5.9
Philippines	8	6.7
Somalia	7	5.9
Sudan	20	16.8
UAE	6	5.0
UK	6	5.0
**Nationality group**		
Arabic	42	35.0
African	15	12.5
European	11	9.2
Asian	52	43.3
**Sex**		
Male	2	1.7
Female	**118**	**98.3**
**Religion**		
Muslim	79	65.8
Christian	26	21.7
Other	15	12.5
**Designation**		
Nurse/Midwife	13	10.8
Resident doctor	9	7.5
Medical officer	9	7.5
Specialist	35	29.2
Consultant	54	45.0
**Place of practice**		
Private clinic	15	12.5
Private hospital	41	34.2
Governmental clinic	9	7.5
Governmental hospital	50	41.7
Other	5	4.1

**Table 2 ijerph-20-04710-t002:** Participants’ practice (attitude) about FGM (*n* = 120).

Question	*n* Frequency (*n*)	Percent (%)
**Should FGM/C be internationally prohibited?**		
Yes	99	82.5
No	21	17.5
**Is that FGM culturally accepted for you?**		
Yes	35	29.2
No	85	70.8
**Is that FGM religiously recommend for you?**		
Yes	9	7.5
No	111	92.5
**If you are going to do it or have done it, would you use anesthesia?**		
Yes	33	27.5
No	8	6.70
Not Applicable	79	65.8
**What is the minimal age (years) you can do it?**		
2	1	0.8
9	1	0.8
14	2	1.7
Not Applicable	116	96.6
**Total Attitude Score = 90.3%: Positive attitude**		

**Table 3 ijerph-20-04710-t003:** Comparisons of the mean score of knowledge between Arabian and other nationality groups, ANOVA: (Scheffe Model).

	F-Test	*p*-Value	Nationality Group		Mean ± Std. Deviation	*p*-Value
Between Groups	3.640	0.015	Arabic M ± SD 74.57 ± 10.6	African	72.47 ± 9.4	1.000
				European	70.55 ± 13.4	0.774
				Asian	66.30 ± 11.7	0.008

The mean difference is significant at the 0.05 level.

## Data Availability

Not applicable.

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
