# Peer review of "Legal Awareness and Practices of Female Genital Mutilation/Cutting (FGM/C) among United Arab Emirates Medical Practitioners"

_ijerph, 2023, doi:10.3390/ijerph20064710_

Round 1

Reviewer 1 Report

This is an outstanding piece of research. I have three questions/comments that you may want to consider for further discussion.

1. Is there a correlation between the nationality of the participant and the laws/customs about FGM/C in that country? Are medical practitioners more or less likely to support or oppose it based on the laws/customs in the country they are from? 

2. Since nearly all of the participants were women, perhaps some note about gender being/not being a significant dependent variable. It seems like there is more variation among women than between women and men (can't draw the second conclusion from this study since nearly all women, but perhaps from other studies).

3. Since there are no legal penalties for carrying out FGM/C in UAE, it seems you may want to emphasize that there were no repercussions for complete honesty of performing or accepting it. I think this is a strength of this study that the responses are more likely to be honest than in places where it is illegal/condemned in other ways. 

Author Response

Dear Reviewer,

At the beginning, we want to thank you for all valuable remarks. We are happy that you enjoyed this topic. The manuscript has been revised according to your suggestions. Detailed answer is provided below.

1.Is there a correlation between the nationality of the participant and the laws/customs about FGM/C in that country? Are medical practitioners more or less likely to support or oppose it based on the laws/customs in the country they are from?

We performer additional statistical analysis and added to the manuscript complimentary information:”Regarding participants’ opinions according to their nationality, there was no statistically significant association (p-value>0.05). However, Europeans reported the highest proportion for “Internationally abandon” (90.9%), while Africans had the highest proportion for “Culturally accepted” (13.3%) and for “Religiously recom-mended” (6.7%). Moreover, 46.7% of Africans reported that the procedure is “Reli-giously recommended” (p-value <0.001).

  1. Since nearly all of the participants were women, perhaps some note about gender being/not being a significant dependent variable. It seems like there is more variation among women than between women and men (can't draw the second conclusion from this study since nearly all women, but perhaps from other studies).

This was not a part of our study  but can be find in ather article added in the manuscript. Since nearly all of the participants were women, we didn’t take into consideration sex of participants as dependent variable.

  1. Since there are no legal penalties for carrying out FGM/C in UAE, it seems you may want to emphasize that there were no repercussions for complete honesty of performing or accepting it. I think this is a strength of this study that the responses are more likely to be honest than in places where it is illegal/condemned in other ways. 

Thank you for your valuable suggestion. The statment has been included in Strengths of the study.

Best regards

Kornelia Zareba and Co-authors

Reviewer 2 Report

The submitted manuscript details the results of a study of the attitudes towards, experiences performing, and knowledge about female genital mutilation among medical professionals in the United Arab Emirates. The manuscript should be of great interest to a wide-range of scholars and members of the general public as it addresses an international human rights issue. The rationale and methodology are clearly presented, and the analyses conducted were appropriate. Overall the writing is easy enough to follow in most places, but I would recommend the Abstract in particular as an area in need of revision for clarity.

Author Response

Dear Reviewer,

At the beginning, we want to thank you for all valuable remarks. We are happy that you enjoyed this topic.  The Abstract has been revised and changed according to your suggestions. Unfortunately due to editorial limitation (200 worlds) we couldn’t include all important information in this section.

Best regards

Kornelia Zareba and Co-authors

Reviewer 3 Report

Introduction needs improvement in English usage. 

Table 2 -- "childbirth" (one word)

Table 3 -- what is "presanting" FGM?  Also, should the question read "What type of complication was it?"

Table 5 -- wrong title/title incorrect.

Should be "i.e.," (add comma)

Line 363 "has" rather than "have"

Line 394 "mental health" rather than "mental life"

Line 409 "which" not "with"

Paper needs a very thorough editing for clarity and conciseness, and to promote readability/understanding by readers.

Author Response

Dear Reviewer,

At the beginning, we want to thank you for all valuable remarks. We are happy that you enjoyed this topic. The manuscript has been revised according to your suggestions. Detailed answer is provided below.

  1. Introduction needs improvement in English usage. 

The introduction has been revised and corrected

  1. Table 2 -- "childbirth" (one word)

The word has been changed according to your suggestions.

  1. Table 3 -- what is "presanting" FGM?  Also, should the question read "What type of complication was it?"

The sentence have been changed on „Difficult to deliver her baby without perineal tears or episiotomy”. The second sentence have been corrected according to your suggestions.

  1. Table 5 -- wrong title/title incorrect.

The title has been changed according to your suggestions.

  1. Should be "i.e.," (add comma)

Have been corrected according to your suggestions.

  1. Line 363 "has" rather than "have"

The word has been corrected according to your suggestions.

  1. Line 394 "mental health" rather than "mental life"

The word has been corrected according to your suggestions.

  1. Line 409 "which" not "with"

The word has been corrected according to your suggestions.

  1. Paper needs a very thorough editing for clarity and conciseness, and to promote readability/understanding by readers.

The article has been revised and corrected by native editor according to your suggestion.

Best regards

Kornelia Zareba and Co-authors

Reviewer 4 Report

Awar et al., did wonderful work on awareness of medical Law and practices of female Genital Mutilation in UAE. The education of the population in terms of sexual life and religious thoughts may change incorrect mental patterns occurring in the social consciousness. This is a very important study for medical professionals to accept the circumcision of women. It raised an important and real scenario, which has not been exposed. Sensitizing the public and medical professionals to the practice, as well as the necessity to pass a law explicitly criminalizing it and the duty to report any cases of female circumcision, should be the primary focuses of future efforts.

The topic is original and many needs to create awareness among medical practitioners, Because, Intercultural mediation is increasingly recognized as a crucial tool in the medical industry, particularly when dealing with culturally and ethically complex challenges. To reach an agreement and provide concrete legislative recommendations that could be enforced by medical professionals and society at large, it appears reasonable to form a committee with a representative from the fields of law, medicine, and religion.

Study control is missing methodology, Authors need to clarify it.

I would suggest including at least one figure to make it more visible to readers.  

Overall, this study is very interesting, I recommend it to publish it.  

Author Response

Dear Reviewer,

At the beginning, we want to thank you for all valuable remarks. We are happy that you enjoyed this topic. The manuscript has been revised according to your suggestions. Detailed answer is provided below.

  1. Study control is missing methodology, Authors need to clarify it.

Due to the cross-sectional character of the study there was no control group. The information has been added in the manuscript.

  1. I would suggest including at least one figure to make it more visible to readers.

Table 2 has been changed on Figure1 and 2.

Best regards

Kornelia Zareba and Co-authors

Round 2

Reviewer 3 Report

The manuscript is essentially acceptable. I recommend the following minor revisions to increase readability and proper, consistent, and most common/accepted use of English grammar:

1. In lines 86, 111, 162, 176, 242, 243, 268, 320, 327, 386-87, the authors use the phrase "the majority." Because "the" is a DEFINITE article, its use is reserved for an instance in which there is a single majority--thus "the" majority. In all instances here, the numbers are just one of many possible majorities, so the proper grammar would be "a majority." (If one wants to see a comparable, PROPER use, read line 328. "A total"--which is CORRECT! "The total" would be incorrect.

2. line 363:  Rather than "the overwhelming majority," use "an overwhelming majority.

3. Similarly, remove "the" in lines 177, 409, 422, and 459.

4. Line 464: "woman" should be "woman's"

5. Finally, replace "as regards" with "regarding" in lines 22, 71, 78, and 118.

Author Response

Dear Reviewer,

At the beginning, we want to thank you for all valuable remarks. We are happy that you enjoyed this topic. The manuscript has been revised according to your suggestions. Detailed answer is provided below.

  1. In lines 86, 111, 162, 176, 242, 243, 268, 320, 327, 386-87, the authors use the phrase "the majority." Because "the" is a DEFINITE article, its use is reserved for an instance in which there is a single majority--thus "the" majority. In all instances here, the numbers are just one of many possible majorities, so the proper grammar would be "a majority." (If one wants to see a comparable, PROPER use, read line 328. "A total"--which is CORRECT! "The total" would be incorrect.

Thank you for your kind suggestions. The phrases have been changed according to your suggestions (highlighted in green).

  1. Line 363: Rather than "the overwhelming majority," use "an overwhelming majority.

The phrase has been changed according to your suggestions (highlighted in green).

  1. Similarly, remove "the" in lines 177, 409, 422, and 459.

The phrases have been changed according to your suggestions (highlighted in green).

  1. Line 464: "woman" should be "woman's"

The phrase has been changed according to your suggestions (highlighted in green).

  1. Finally, replace "as regards" with "regarding" in lines 22, 71, 78, and 118.

The phrases have been changed according to your suggestions (highlighted in green).

Best regards

Kornelia Zareba and Co-authors